# Quantitative Analysis of Drag Force for Task-Specific Micromachine at Low Reynolds Numbers

**DOI:** 10.3390/mi13071134

**Published:** 2022-07-18

**Authors:** Qiang Wang, Zhen Wang

**Affiliations:** 1Infrastructure Management Department, Wuhan University of Technology, Wuhan 430070, China; qiang_wang@whut.edu.cn; 2Hubei Key Laboratory of Theory and Application of Advanced Materials Mechanics, Department of Mechanics and Engineering Structure, Wuhan University of Technology, Wuhan 430070, China

**Keywords:** conical micromotor, hydromechanics, Navier–Stokes equation, drag force

## Abstract

Micromotors have spread widely in order to meet the needs of new applications, including cell operation, drug delivery, biosensing, precise surgery and environmental decontamination, due to their small size, low energy consumption and large propelling power, especially the newly designed multifunctional micromotors that combine many extra shape features in one device. Features such as rod-like receptors, dendritic biosensors and ball-like catalyzing enzymes are added to the outer surface of the tubular micromotor during fabrication to perform their special mission. However, the structural optimization of motion performance is still unclear. The main factor restricting the motion performance of the micromotors is the drag forces. The complex geometry of a micromotor makes its dynamic behavior more complicated in a fluid environment. This study aimed to design the optimum structure of tubular micromotors with minimum drag forces and obtain the magnitude of drag forces considering both the internal and external fluids of the micromotors. By using the computational fluid dynamics software Fluent 18.0 (ANSYS), the drag force and the drag coefficient of different conical micromotors were calculated. Moreover, the influence of the Reynolds numbers Re, the semi-cone angle δ and the ratios ξ and η on the drag coefficient was analyzed. The results show the drag force monotonically increased with Reynolds numbers Re and the ratio η. The extreme point of the drag curve is reached when the semi-cone angle δ is 8° and the ratio ξ is 3.846. This work provides theoretical support and guidance for optimizing the design and development of conical micromotors.

## 1. Introduction

Over the past 20 years, the field of micromachines has developed rapidly with many teams from around the globe. Efficient and fast micromotors, based on various propulsion mechanisms and geometries and materials, have been applied to a wide range of biomedical applications. Traditional synthetic micromachines have been shown to perform well in biological media [1,2,3], large cargo delivery [4,5,6,7,8], precise microsurgery [9,10,11], cell biosensing [12,13,14,15] and environmental decontamination [16,17,18,19,20]. In order to meet the demands of different specific applications, various geometries of micromotors with their propulsion mechanisms have been proposed. Bubble-propelled catalytic microjets, which convert chemical energy into kinetic energy, display high speed and efficiency [21,22,23]. Based on Li’s experiments [24], conical micromotors have higher propulsion efficiency than other motors, including Janus microspheres [25,26], rod micromotors [27,28], nanowires [29], nanoshell micromotors [30] and 3D print heterotypic structures [31,32].

Up to now, bubble-driven tubular micromotors have generated a higher propulsion force than other geometries. The tubular micromotors can move faster than others since there are two kinds of forces influencing the motion of micromotors. One is the driving force, and the other one is the drag force caused by viscosity and pressure of the flow field [33,34]. The driving force comes from the bubbles nucleating from the decomposition of fuel fluid into gas due to the catalyst Pt layer. According to Klausner [35], the driving force depends on the fluid viscosity of solution, the gas productivity speed and the radius of the bubble. These parameters usually can be gained from experiments. However, the drag force is difficult to measure from experiments since the size of the micromotor is too small. A simplified method of calculating the drag force was proposed by Cox [36]. This method was first proposed to describe the drag force of an ellipsoid, and then researchers used it to calculate the drag force on a circular cylinder of finite length and a long spheroid [22,33]. After that, a modified drag force formula was proposed by Li [24]. Complex modified parameters are introduced to describe the drag force of conical micromotors based on the original equations mentioned above [37]. All the modified formulas are used to determine the drag force on the tubular micromotors. All the researchers neglect one main problem: This formula is used to calculate the ellipsoid, which only has outer surfaces. However, the tubular micromotor has an inner face, and the drag force caused by the inner face cannot be ignored compared to other faces.

Considering the major challenges of specific applications in the future, more advanced micromotors, combining multiple functions, will be created to meet the needs of complex biomedical tasks. As shown in Figure 1, researchers have turned their attention from single-task micromotors (A) to multifunctional micromotors (B) [38]. Various new functionalities and capabilities have been added to the tubular micromotors, such as enzyme, antigen and antidote. These sensing devices made the outer surface of the tubular micromotors not smooth anymore. Hence, the drag force becomes more complicated, especially when the Reynolds number is low, and the viscous force, caused by the shearing motion of the fluid, plays a major part in drag force [39]. Fluid resistance is dependent upon the physical properties of fluids, the geometric parameters of micromotors [40] and the motion of fluids.

In this paper, the hydrodynamics theory is applied to the calculation in order to simulate the drag force of the general-purpose micromotor. Navier–Stokes equations and the continuity equation are established for the surrounding flow field [41]. The ANSYS Fluent solver is used to execute computational fluid dynamics (CFD) simulations and calculate the drag force [42]. An unstructured mesh was used for all simulations, and mesh independence studies were carried out to ensure that the final CFD solution was free of mesh resolution errors [43]. This paper aims to investigate the drag force of unsmooth conical micromotors. By using the normalization method, we try to investigate the motion of the general-purpose micromotor.

## 2. Theory and Method

We use a simplified model to simulate the various sensing structures and the tubular micromotor immersed in the fluid field as shown in Figure 2a. The geometries of the sensors varied from each other; in this paper, we assume the sensors are all hemispheres on the outer surface (Figure 2b). The convex hemispheres are randomly distributed on the outer surface of the micromotor, in order to simulate the influence of drag force caused by sensors.

In this paper, the micromotor moves at a very low Reynolds number since the size of the micromotor is so small. The Reynolds number (*R_e_*) is the ratio of inertial forces to viscous forces, which can be used to predict flow patterns in different flow situations. It can be defined as $R_{e}=\rho vL/\mu$. Here, *ρ* is the density of the fluid, *v* indicates the average velocity of a micromotor, *L* = 2*R*_max_ is the larger diameter of the micromotor and *μ* is the dynamic viscosity of the fluid. Thus, the viscous resistance is remarkable, which causes a drag force to be applied to the micromotor as it moves in the fluid.

To calculate the drag force of a conical micromotor, a cylindrical coordinate system (r, θ, x) is established as follows: The *X*-axis is along the length of the micromotor. The parameters *L*, *R*_max_ and *V*_∞_ denote the length, the larger radius of the micromotor and the fluid velocity distance from the micromotor. According to the Navier–Stokes and general continuum equations, the relationship between the pressure and velocity of fluid around the micromotor can be described as follows:(1)$$\{\begin{array}{c} \frac{V_{r}}{r}+\frac{\partial V_{r}}{\partial r}+\frac{\partial V_{x}}{\partial x}=0\\ \frac{\partial P}{\partial r}=\mu \left(\frac{1}{r}\frac{\partial }{\partial r}\left(r\frac{\partial V_{r}}{\partial r}\right)+\frac{\partial^{2}V_{r}}{\partial x^{2}}-\frac{V_{r}}{r^{2}}\right)\\ \frac{\partial P}{\partial x}=\mu \left(\frac{1}{r}\frac{\partial }{\partial r}\left(r\frac{\partial V_{x}}{\partial r}\right)+\frac{\partial^{2}V_{x}}{\partial x^{2}}\right) \end{array}$$
where *V_r_* is the speed of the flow field in the *r* direction, *V_x_* is the speed of the flow field in the *x* direction, *μ* is the dynamic viscosity of fluid and *P* is the pressure of the fluid. As the Reynolds number is relatively low, the inertial force and gravity of fluid can be neglected.

Even though the boundary conditions are added to Equation (1), the pressure distribution of the micromotor can be gained from the velocity of fluid:(2)$$\{\begin{array}{l} P_{rr}=-P+2\mu \frac{\partial V_{r}}{\partial r}\\ P_{xx}=-P+2\mu \frac{\partial V_{x}}{\partial x}\\ P_{xr}=\mu \left(\frac{\partial V_{r}}{\partial r}+\frac{\partial V_{x}}{\partial x}\right) \end{array}$$

The drag force *F_drag_* can be obtained by integrating pressure distributions at all the surfaces of the micromotor theoretically. However, for the micromotor with a convex surface shown in Figure 1, there is no analytical solution for Equation (2); that is to say, we cannot get the drag force from Equation (3).
(3)$$F_{drag}=\int_{\Omega }^{}\left(P_{xx}+P_{xr}\right)d\Omega$$
where Ω is the surface of the micromotor. The thickness of the micromotor is ignored since it is much smaller than the characteristic diameter.

In this paper, we choose the CFD method to simulate the drag force since the theoretical and experimental methods all failed. The Π-theorem is a commonly used theorem in dimensional analysis [44], especially in solving the drag force of a solid in a fluid. Usually, *ρ*, *v* and D are selected as the basic physical parameters. According to the Π-theorem, five independent Π numbers are obtained, namely drag coefficient *C_d_*, the Reynolds number *R_e_*, the semi-cone angle δ, the ratio of length to opening diameter ξ and the ratio of convexities on the outer surface η. η is the ratio of the total surface area of the convex part to the smooth outer surface (Figure 2 shows η = 50% as an example). If the number of convex hemispheres is m, then the ratio η is given by Equation (4).
(4)$$\eta =\frac{m\cdot 2\pi r^{2}}{\pi \left(R_{max}^{2}-R_{min}^{2}\right)/\tan\delta }=\frac{2mr^{2}\tan\delta }{R_{max}^{2}-R_{min}^{2}}$$
where *r* is the radius of the convex hemisphere on the outer surface, $R_{max}^{}$ and $R_{min}^{}$ are the radiuses of the openings at both ends for a tubular micromotor.

Based upon the dimensional analysis, the relationship is given by Equation (5).
(5)$$C_{d}=\frac{F_{drag}}{\frac{1}{2}\rho Av^{2}}=f\left(R_{e},\delta ,\xi ,\eta \right)$$

The reference area A is the frontal area of a micromotor on a plane, perpendicular to the flow direction, which is expressed as follows:(6)$$A=\pi \left(R_{max}^{2}-{\left(R_{max}-L\tan\delta \right)}^{2}\right)$$

The radius of the bigger opening is 20.0 μm, whereas the length of the micromotor is 100.0 μm. Moreover, the thickness of the tubular micromotor is 1.0 μm. The fluid medium was water with a density of 998.2 kg/m^3^ and a viscosity of 1.003 mPa·s. This paper aimed to simulate the relationship between the drag force and the influence factors. According to its definition, the Reynolds number mainly depends on the velocity when the properties of the fluid are fixed. Therefore, the speed was within the range of 0.02–10 mm/s.

The computational domain and boundary conditions used in Fluent are shown in Figure 3. The micromotor seems to be very small compared to the fluid surroundings. That is to say, fluid is infinite in comparison to the objects moving in it. Thus, the left side is the velocity inlet boundary, the right side is the pressure outlet boundary, and the other sides are the outflow boundaries, in order to simulate the micromotors moving in an infinite fluid. The micromotor was immersed in a fluid. The density of the fluid is 998.2 kg/m^3^, and the dynamic viscosity is 1.003 mPa·s. The laminar flow model was chosen as the Reynolds number is within a small range.

## 3. Results and Discussion

The drag force of the micromotor is calculated by integrating the pressure on the micromotor surfaces in the flow direction (Figure 4a). Figure 4b shows how the fluid flows over the surfaces of the micromotor. The velocity of the fluid slows down when the tube is on the way there. According to the dimensional analysis method, the simulation models are divided into four groups; each group calculates the influencing factor separately.

### 3.1. Reynolds Number (R_e_)

According to the numerical simulation results, the relationship between the drag forces and drag coefficient under different Reynolds numbers is shown in Figure 5. The Reynolds numbers ranging from 4 × 10^−^^3^ to 0.2 are presented below. As shown in Figure 5a, the drag force of a conical micromotor increases with the increase in the Reynolds number. According to the definition of the Reynolds number $R_{e}=\rho vL/\mu$, when the fluid is chosen, the density and viscosity of the fluid are fixed. So, the Reynolds number is only related to the fluid velocity. It has been previously found that for a smooth tubular micromotor, the drag forces of the micromotor increased linearly with the increase in the Reynolds number [39]. However, here, for the tubular micromotor with convex shapes on the outer surface, the drag force increases monotonically but no longer linearly. On the contrary, as pointed out in Figure 5b, the drag coefficient decreases as the Reynolds number increases. The results highlight the different dependencies on the Reynolds number between the drag force and the drag coefficient.

### 3.2. The Semi-Cone Angle (δ)

Considering different semi-cone angles of conical micromotors ranging from 1° to 11°, different models are calculated. We assume that the small opening of the tubular micromotor R_min_ and the length L are fixed, so the big opening R_max_ varies while the semi-cone angle changes from 1° to 11°. There is a local minimum drag force when the semi-cone angle increases, as shown in Figure 6a, when the semi-cone angle is 8°. The drag coefficient for the conical micromotor also decreases with the increase in a semi-cone angle. The same conclusion has been given in light of Li’s experimental results [24]. However, as shown in Figure 6b, the slope indicating the relationship between the drag coefficient and the semi-cone angle becomes smaller and smaller, indicating that the semi-cone angle has a greater impact on the drag coefficient when it is small.

In fluid dynamics, Equation (3) is a formula used to calculate the force of drag experienced by an object due to movement through a fully enclosing fluid. The drag force usually consists of both a skin friction component and a form drag component (also known as pressure drag force). The normal stress on the surface of the tubular micromotor contributes to the form drag, which is why it is also called the pressure drag. The shear stress on the surface of the tubular micromotor contributed to the skin friction drag. For smooth bodies, such as a cylinder, the skin friction force may become significant when Reynolds numbers are small. For sharp-cornered bluff bodies, such as square cylinders and plates held transverse to the flow direction, the form drag force plays an important role in the whole drag force when the Reynolds number is large. That is to say, when the Reynolds number is very small, the skin friction force plays the major role; otherwise, the form drag force plays the dominant role.

In this paper, the Reynolds number is smaller than 1, so in order to analyze the skin friction force among the micromotor, we simulate the force on each surface of the motor. As shown in Figure 7, motor_1 is the circular ring area that first faces the fluid flow, motor_2 is the circular ring area at the big opening end of the micromotor, motor_3 is the inner surface of the tube and motor_4 is the outer surface that contains numerous convex shapes.

The results show that the motor_1 and motor_2 surfaces have almost no influence on both skin friction force and form drag force. The inner (motor_3) and outer (motor_4) surfaces play the main role in skin friction force in Figure 8a. In particular, the outer surface accounts for the majority of the skin friction force. As the fluid flows over the micromotor, the surface shear stress occurs on the inner and outer surfaces of the micromotor, and then it applies frictional forces to the surface of the motor which works to impede the forward movement of the motor. The total skin friction force in Figure 8a increases while the total form drag force in Figure 8a decreases; that is the reason why the total drag force has a threshold value in Figure 6a.

### 3.3. The ratio of Length to Opening Diameter (ξ)

Similarly, Figure 9a shows the relationship between the drag forces and the parameter ξ. The relationship between the drag forces and ratio ξ was found not to be monotonous. The drag force was minimal when the value of ξ lay within the range of 3.5–4.0. Both the drag force and drag coefficient for conical micromotor decrease with the ratio increase. When the ratio ξ increases from 1.25 to 5.0, the length increases while the larger radius remains unchanged. At the same time, the radius of the small opening decreases as the length increases. The drag force decreases with the decrease in the smaller radius and the increase in length. There is also a local minimum drag force when the ratio of length to opening diameter ξ increases as shown in Figure 9a, when the ratio ξ is 3.846. So, the drag force is very sensitive to geometry when the fluid flows in a low velocity range. Thus, more attention should be paid to the geometry design in order to obtain more efficient micromotors in this velocity range.

### 3.4. The Ratio of Convexities on the Outer Surface (η)

We use parameter η to describe the quantity of convexities on the outer surface of the micromotor by Equation (4). Thus, the drag force increases while the number of convexities increases on the outer surface. The surface area of the outer surface increases while the parameter *η* increases, but the projected area of the micromotor remains unchanged when it flows over the fluid. So it is found that the trend of graphs in Figure 10a,b is consistent. When the parameter *η* is less than 50%, the drag force is relatively stable. However, when *η* exceeds 50%, the drag force increases rapidly. So, few sensors installed on the outer surface of the motor have little effect on its motion.

According to the results, the drag coefficient of a micromotor decreases nonlinearly, along with the increase in the Reynolds number, semi-cone angle and ratio of length to larger radius. However, the drag coefficient increases along with the increase in the ratio of convexities on the outer surface. These figures demonstrate how geometry and flow field influence the drag force acting on the micromotors. Obviously, the drag coefficient and geometric parameters are nonlinear relationships, and the analyzed parameters above are coupled with each other. Through a data-fitting method and analysis, a certain relationship among dimensionless quantities will be obtained in the future.

## 4. Conclusions

Considering the advanced fabrication and functionalization technologies, multifunctional micromotors are capable of performing diverse tasks. However, despite these tremendous technological advances, it is extremely challenging to investigate the motion of micromotors with abnormal shapes. Typical analytical methods for the drag force cannot be adapted to an abnormally shaped motor since the Navier–Stokes equations have no analytic solution based on hydrodynamic theory. In this paper, a numerical simulation was introduced to solve the inhomogeneous partial differential equations. A numerical model used to describe the relationship between dimensionless quantities, including *C_d_*, *R_e_*, *δ*, *ξ* and *η*, has been built. The results showed that the drag force increases nonlinearly with the increase in the Reynolds number *R_e_*. However, the drag coefficient decreases nonlinearly as the Reynolds number increases. Meanwhile, the drag force has a threshold value when the semi-cone angle *δ* increases, while the drag coefficient decreases nonlinearly. Furthermore, both the drag force and the drag coefficient decrease nonlinearly with the increase in the ratio *ξ* while both the drag force and the drag coefficient increase nonlinearly with the increase in the ratio *η*. Compared with the two local minimum drag forces from Figure 6a and Figure 9a, when the ratio ξ is 3.846, the drag force reaches a smaller value. So, the optimal geometry with minimum fluid drag force is as follows: the radius of the bigger opening is 20.0 μm, the semi-cone angle is 5° and the length of the micromotor is 153.84 μm.

However, some key problems still remain for the motion of micromotors, such as the bubbles, and it is suggested that the behavior of bubbles will need to be considered in the calculation of the drag force of the micromotor. For example, when a bubble nucleates and grows on the inner surface of the tubular micromotor, the fluid field changes rapidly when the bubble blows off the liquid around it. Moreover, there may be mountains of bubbles that exist at the same time since the fuel solution reacts vigorously with catalysts in it. Furthermore, as the chemical reactions carry on, a concentration difference develops along the surface of the motor, which generates a pressure gradient. The pressure gradient also influences the fluid field around the bubbles, which is normally called self-diffusiophoresis. Since the repulsive interaction is weak, self-diffusiophoresis was shown to diminish when the critical size of the motor is not very small. However, at present, these factors are not often considered. This shows that putting too many variables into a model ends up degrading the results. Perhaps we will take the bubbles and self-diffusiophoresis into consideration in our future research.

It is clear that realizing the vision of intelligent micromotors and expanding their scope require the close collaboration of researchers in diverse fields; we hope our research can help to assess and address the most pertinent challenges.

## Figures and Tables

**Figure 1 micromachines-13-01134-f001:**
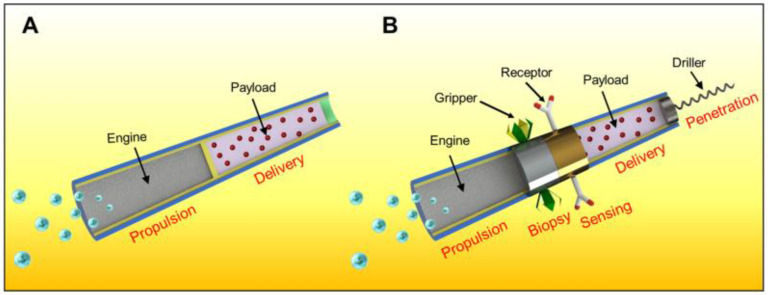
Schematic of the micromotors [38]: (**A**) special-purpose micromotor; (**B**) general-purpose micromotor.

**Figure 2 micromachines-13-01134-f002:**
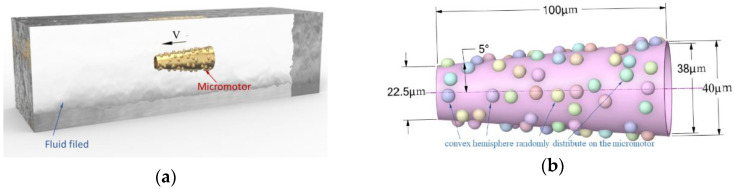
Schematic of the micromotor: (**a**) micromotor immersed in the fluid; (**b**) geometry of the micromotor.

**Figure 3 micromachines-13-01134-f003:**
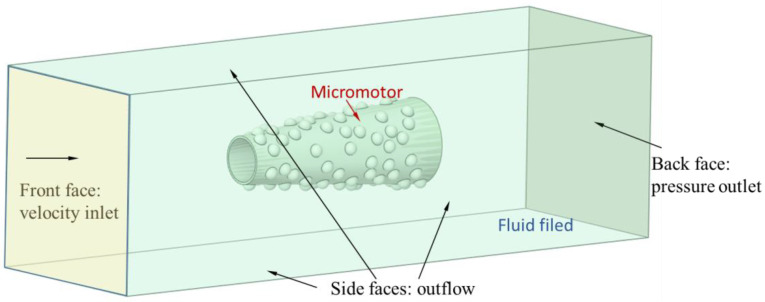
Numerical model and boundaries for simulating the drag force.

**Figure 4 micromachines-13-01134-f004:**
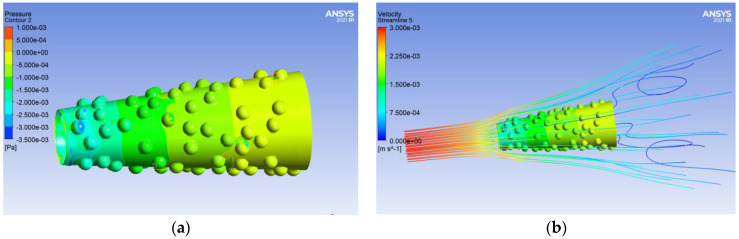
Results calculated by Fluent numerical calculation software. (**a**) The pressure distribution on the surface of the micromotor; (**b**) the velocity distribution of the flow field around the micromotor.

**Figure 5 micromachines-13-01134-f005:**
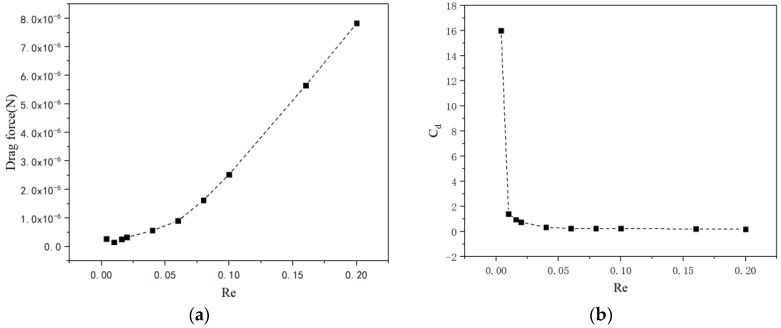
The drag force (**a**) and drag coefficient (**b**) versus the Reynolds number ranging from 4 × 10^−^^3^ to 0.2.

**Figure 6 micromachines-13-01134-f006:**
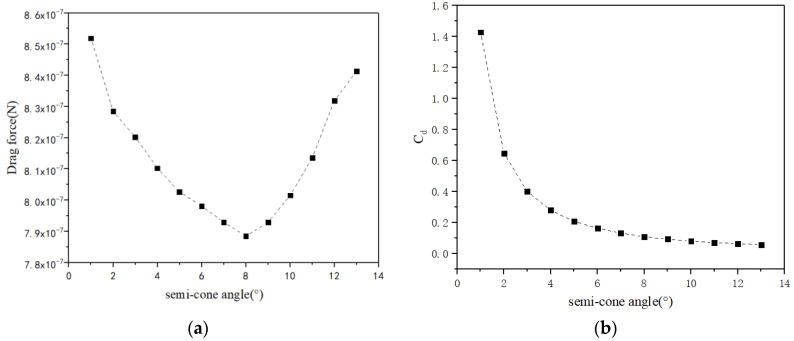
The drag force (**a**) and drag coefficient (**b**) versus semi-cone angle ranging from 1° to 7°.

**Figure 7 micromachines-13-01134-f007:**
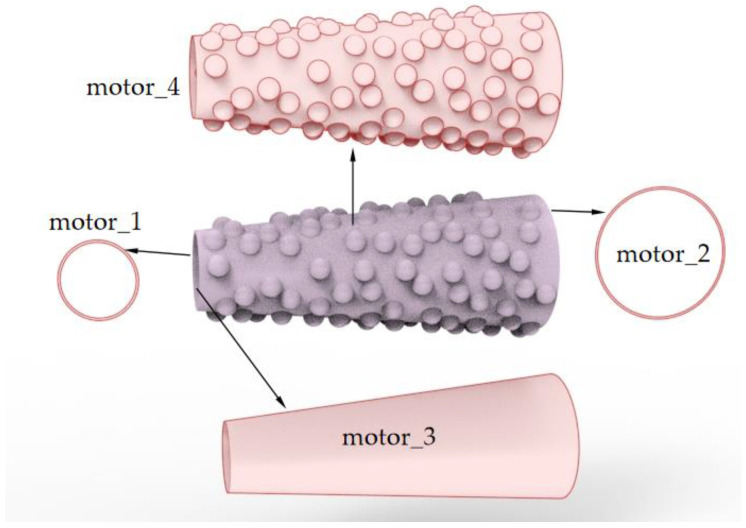
Schematic of the faces attached to the tubular micromotor.

**Figure 8 micromachines-13-01134-f008:**
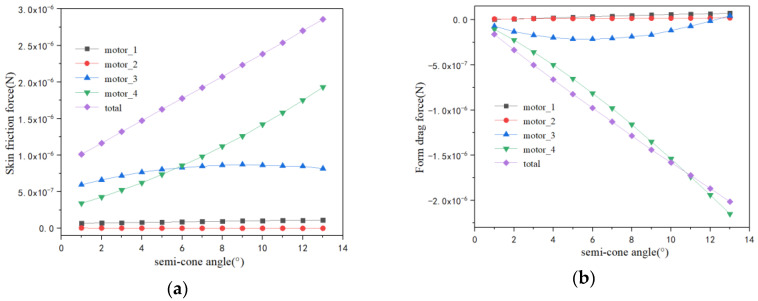
The drag force (**a**) and drag coefficient (**b**) versus semi-cone angle ranging from 1° to 7°.

**Figure 9 micromachines-13-01134-f009:**
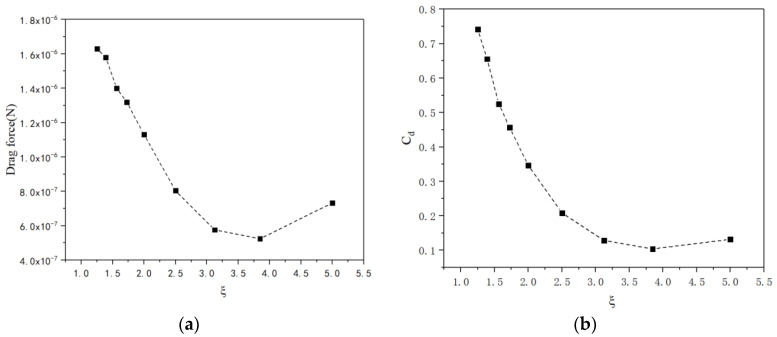
The drag force (**a**) and drag coefficient (**b**) versus the rate of length to larger diameter ranging from 1.25 to 5.

**Figure 10 micromachines-13-01134-f010:**
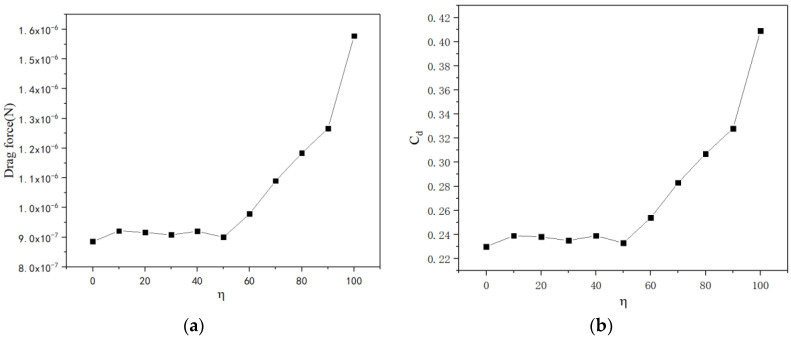
The drag force (**a**) and drag coefficient (**b**) versus the ratio of length to larger radius ranging from 4 to 7.

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
