# Peer review of "Quantitative Analysis of Drag Force for Task-Specific Micromachine at Low Reynolds Numbers"

_micromachines, 2022, doi:10.3390/mi13071134_

Round 1
Reviewer 1 Report
The authors report a CFD model to calculate the drag force of catalytical tubular micromotors with various geometrical parameters. To optimize the geometrical design of the tubular motors, the authors calculated drag forces of motors with various length, radius, surface roughness, cone angle within low Reynolds number regime.
The authors need to address the following critical question before this paper can be considered to be publishable:
This CFD model assumes liquid filled in the entire inner cavity of the motor. However, the motor is driven by bubbles generated inside the tube, and propelled by the bubble detachment at the opening end. Furthermore, the catalytical reaction is also affected by the geometry of the motor. For instance, diffusion of the chemical fuel can be slow for a motor with small radius of opening. These huge differences between the actual experimental scenario and the simulation assumptions makes those results highly questionable, and I am not convinced unless there is additional evidence proving that the simplification of the model is acceptable.
Author Response
Please see the attaches below.

Reviewer 2 Report
In this study, the authors show quantitative analysis of drag force for task-specific micromachine at low reynolds numbers. This work is interesting and inspiring, and is well written. However, a minor revision is still required before acceptance for publication in Micromachines.
(1) This study aims to design the optimum structure of tubular micromotors with minimum drag forces. Accordingly, a direct conclusion about the optimum structure should be given in Abstrace, Results and Discusion, and Conclusion parts.
(2) In Figure 2, the diagram of the micromotors is too simple, and its structure should be clearly marked and give detailed demonstration to enhance the readability. And the same for Figure 3.
Author Response
Please see the attaches below.

Round 2
Reviewer 1 Report
The authors address my question about the validity of this model. This numerical model takes both the drag forces from outer and inner surfaces into consideration and improves the accuracy of drag calculation compared to the Cox model that was commonly used in the field. I agree that this model could lead to a better understanding of the dynamics of microtubular motors, as well as the optimization of the motor design from a hydrodynamic perspective for the community.
The author also mentioned the potential influence of bubbles inside the cavity. The effect of bubbles could be beyond the scope of this paper, but a brief discussion should be included in the manuscript.
As for the discussion about chemical reactions, the diffusion of chemical fuels is not a negligible factor. It highly depends on the dimension of the motor. Just think about an extreme case, i.e., a motor with a very small opening, the flux of incoming fuels is limited, and the fuel concentration can be lower than the bulk concentration as the motor moves. The diffusion and convection of chemical fuels should at least be discussed, if not simulated.
Author Response
We have already added the contents about the influence of bubbles and self-diffusiophoresis in the conclusion part in the revised manuscirpt. The contents are as follows:
However, some key problems still remain for the motion of micromotors such as the bubbles, and it is suggested that the drag force of micomotor will need to consider the be-havior of bubbles. For example, when the bubble nucleates and grows up at the inner sur-face of the tubular micromotor, the fluid filed changes rapidly when the bubble blows off the liquid around it. Moreover, there may be mountains of bubbles exist at the same time since the fuel solution reacts vigorously with catalysts in it. Furthermore, as the chemical reactions carries on, a concentration difference develops along the surface of the motor, which generates a pressure gradient. The pressure gradient also influences the fluid field around the bubbles, which is normally called self-diffusiophoresis. Since the repulsive in-teraction is weak, self-diffusiophoresis was shown to diminish when the critical size of motor is not very small. However, at present these factors are not often considered. It shows putting too many variables into a model ends up degrading the results. Maybe we will take the bubbles and self-diffusiophoresis into consideration in our future research.
